# Variability and performance of NHS England's 'reason to reside' criteria in predicting hospital discharge in acute hospitals in England: a retrospective, observational cohort study

Elizabeth Sapey [1,2] Suzy Gallier,[1,3] Felicity Evison [3] David McNulty,[3] Katherine Reeves,[3] Simon Ball [4,5]

ES and SG are joint first authors.

For numbered affiliations see end of article.

**Correspondence to**
Dr Elizabeth Sapey;
e.sapey@bham.ac.uk

## ABSTRACT

**Objectives** NHS England (NHSE) advocates 'reason to reside' (R2R) criteria to support discharge planning. The proportion of patients without R2R and their rate of discharge are reported daily by acute hospitals in England. R2R has no interoperable standardised data model (SDM), and its performance has not been validated. We aimed to understand the degree of intercentre and intracentre variation in R2R-related metrics reported to NHSE, define an SDM implemented within a single centre Electronic Health Record to generate an electronic R2R (eR2R) and evaluate its performance in predicting subsequent discharge.

**Design** Retrospective observational cohort study using routinely collected health data.

**Setting** 122 NHS Trusts in England for national reporting and an acute hospital in England for local reporting.

**Participants** 6 602 706 patient-days were analysed using 3-month national data and 1 039 592 patient-days, using 3-year single centre data.

**Main outcome measures** Variability in R2R-related metrics reported to NHSE. Performance of eR2R in predicting discharge within 24 hours.

**Results** There were high levels of intracentre and intercentre variability in R2R-related metrics (p<0.0001) but not in eR2R. Informedness of eR2R for discharge within 24 hours was low (J-statistic 0.09–0.12 across three consecutive years). In those remaining in hospital without eR2R, 61.2% met eR2R criteria on subsequent days (76% within 24 hours), most commonly due to increased NEWS2 (21.9%) or intravenous therapy administration (32.8%).

**Conclusions** Reported R2R metrics are highly variable between and within acute Trusts in England. Although case-mix or community care provision may account for some variability, the absence of a SDM prevents standardised reporting. Following the development of a SDM in one acute Trust, the variability reduced. However, the performance of eR2R was poor, prone to change even when negative and unable to meaningfully contribute to discharge planning.

## STRENGTHS AND LIMITATIONS OF THIS STUDY

⇒ The intracentre and intercentre variability of reason to reside (R2R) reporting was based on national data and included >6.6M patient-bed-days.
⇒ Standardised data model to form eR2R was based on nationally agreed criteria for each clinical question.
⇒ All admissions >24 hours were included for eR2R performance review, reducing bias.
⇒ eR2R data based on one centre only, although one of the largest National Health Service Trusts nationally serving a diverse population and including >1M patient bed-days.

## INTRODUCTION

In 2021, the UK Government published its policy and operating model for hospital discharge and community support within the National Health Service in England (NHSE).[1] This policy responded to concerns about bed capacity during the COVID-19 pandemic.

A National Audit Office report recognised the potential to release acute hospital beds in 2016, finding that older patients no longer needing acute treatment accounted for 2.7 million NHS hospital bed days per year.[2] The report concluded that a lack of planning delayed discharge, recognising research that highlighted adverse outcomes during prolonged hospital stay.[3 4]

The aforementioned policy mandates using set criteria to identify in-patients in whom discharge home, or to a less acute setting, should be considered. These criteria have been referred to interchangeably, as 'Reason[s] to reside' (R2R), 'right to remain' or 'criteria to reside' (see table 1A). Since April 2020, NHS hospitals have been required to provide daily reports on the numbers of people leaving hospital, to where and the

| | Table 1 Reason to reside (R2R) |
|---|---|
| 1 | Requiring Intensive Care (ITU) or High Dependency Unit (HDU) care |
| 2 | Requiring oxygen therapy/ Non Invasive Ventilation (NIV) |
| 3 | Requiring intravenous fluids |
| 4 | National Early Warning Score (NEWS) 2 >3 (clinical judgement required in persons with Atrial Fibrilliation (AF) and/or chronic respiratory disease) |
| 5 | Diminished level of consciousness where recovery realistic |
| 6 | Acute functional impairment in excess of home/ community care provision |
| 7 | Last hours of life |
| 8 | Requiring intravenous medication more than twice daily (BD) (including analgesia) |
| 9 | Undergone lower limb surgery within 48 hours |
| 10 | Undergone throrax-abdominal/pelvic surgery within 72 hours |
| 11 | Within 24 hours of an invasive procedur (with attendant risk of acute life-threatening deterioration) |

The policy and operating model for hospital discharge and community support within the National Health Service in England states that every person on every general ward should be reviewed on a twice daily ward round to determine whether they meet R2R. If the answer to each question is 'no', the policy states that active consideration for discharge to a less acute setting must be made.[1] In daily data returns, the number of patients to whom this applied were counted at a single, locally defined, time point.

reasons for those remaining in hospital. The proportion of in-patients not meeting R2R criteria and the proportion of patients without R2R discharged that day are also reported. These metrics are considered to be measures of organisational efficiency.

R2R appears to have emerged heuristically from the clinical experience of those involved in its development. A series of questions are posed that might prompt consideration of individual patients for discharge. However, there are no standardised data definitions, there has been no validation of R2R, no investigation of its role as a clinical decision support tool or of its value in evaluating hospital performance. A further barrier to evaluating the performance of R2R is that there is no gold standard definition that identifies patients who could be discharged from hospital against which to compare R2R performance. This lack of a reference standard limits, but does not preclude assessment of the validity of a clinical test, provided a 'fair' measure of performance can be defined.[5] The set of patients actually discharged in the subsequent 24 hours is one potentially 'fair' test of performance of R2R.

In the current study, we show the degree of variation in R2R-associated metrics reported across centres in England. Second, we propose precisely defined, interoperable data definitions corresponding to the elements of R2R. This allows for consistent, generalisable analysis. Third, we evaluate the performance of R2R to predict discharge over the subsequent 24 hours.

## METHODS
All studies activities followed the World Medical Association's Declaration of Helsinki. The R2R criteria are as described[1] and are also provided in table 1.

### National data
National NHS England data were accessed via The UK Health Facts and Dimensions database[6] for all reporting Trusts in England. Assessment of variability in national R2R reporting included data from 29 November 2021 to 20 February 2022. Online supplemental table S1 provides the names of the Trusts whose data are presented anonymously. Data were collected daily during the censor period for 121 centres, yielding a total of 10 164 potential data points (centre-days). For each of these, the total number of occupied and unoccupied beds, and the number of patients with no right to reside were extracted. The number of patients with no right to reside were submitted once a day by each NHS trust, based on the local hospital interpretation of the definition provided by NHSE.[1] This required none of the criteria to be met at the time of local data collection. The numbers of patients with right to reside were then calculated by subtracting the number with no right to reside from the total number of occupied beds on that day. The number of general and acute beds occupied in any given centre, on any given day (in-patients), was used as a surrogate for the number of patients eligible for evaluation using the R2R criteria. Review of the dataset found some missing and potentially spurious data, which were excluded prior to analysis. This included instances where R2R data were not recorded (n=184 data points), where the total numbers of beds were either zero, missing or clearly spurious (n=37 data points) or where there were more patients with no R2R than the total number of beds (n=3 data points). The national data are shown for the other n=121 centres, excluding UHB.

### Local data
In-depth analysis of R2R criteria were performed using data from the Queen Elizabeth Hospital Birmingham (QEHB). QEHB is an NHS, urban, adult, acute hospital in England, which in 2019 had 1269 beds including 80 level 2/3 intensive care unit (ICU) beds, an emergency department that assesses >300 patients per day and a mixed secondary and tertiary practice that includes all major adult specialities except for obstetrics and gynaecology. The electronic healthcare record (EHR) at QEHB (PICS, Birmingham Systems) contains time-stamped, structured records that include demography, location, admission and discharge, comorbidities, physiological measurements supporting NEWS2 and Glasgow Coma Scale, operation noting, prescribing and investigations.

**Table 2** Data definitions used to operationalise R2R for EHR

|  | Flag if… | R2R criterion number |
|---|---|---|
| On ITU HDU | listed as being in ITU or HDU ward | 1 |
| SNCT level ≥2 | Most recent SNCT level in previous 48 hours ≥2 | 1 |
| SNCT level a | Most recent SNCT level in previous 48 hours=1a | 6 |
| SNCT level 1b | Most recent SNCT level in previous 48 hours=1b | 6 |
| Oxygen therapy/NIV | Oxygen administration or Non Invasive Ventilation (NIV) documented in observation chart within previous 24 hours | 2 |
| Intravenous fluids | Intravenous fluid administration initiated in previous 24 hours or variable rate insulin infusion administered in previous 24 hours | 3 |
| NEWS2 | If NEWS2>3 within last 24 hours | 4 |
| Diminished consciousness | Glasgow Coma Scale value ≤12 in last 24 hours | 5 |
| Last hours of life | Comfort observation completed current OR end-of-life medication bundle administered within last 24 hours | 7 |
| Intravenous prescription tds current (regular not prn) | Intravenous medication prescribed within last 24 hours and frequency ≥3 times per day for regular medication only | 8 |
| Intravenous medication administration tds within 24 hours | Intravenous medication administered ≥3 times within last 24 hours | 8 |
| Lower limb surgery within 48 hours | Procedure with relevant OPCS codes in previous 48 hours | 9 |
| Thorax-abdominal-pelvic surgery with 72 hours | Procedure with OPCS relevant codes in previous 72 hours | 10 |
| Invasive procedure within 24 hours | Procedure with OPCS relevant codes in previous 24 hours | 11 |

The table describes the data definitions used and the R2R criteria they map to.
All OPCS codes used to identify procedures are listed in online supplemental table S2.
EHR, electronic healthcare record; HDU, high dependency unit; ITU, intensive care; NEWS2, National Early Warning Score 2; OPCS, OPCS Classification of Interventions and Procedures code, which is used to identify the coded clinical entry; R2R, reason to reside; SNCT, Safer Nursing Care Tool; tds, three times a day.

The R2R criteria in table 1 were mapped to computable definitions derived from the EHR (see table 2), to generate an electronic R2R (eR2R). The OPCS Classification of Interventions and Procedures codes mapped to criteria 9–11 are described in online supplemental table S2. The concept 'acute functional impairment in excess of home/community care provision' had no direct correlate. Safer Nursing Care Tool (SNCT) levels of care were however available.[7] SNCT levels 2 and 3 correspond closely with the requirement for HDU or ICU.[8] Level 1a identifies patients requiring enhanced nursing reflecting acuity of illness, and level 1b identifies a group with increased nursing dependency. Level 1b is likely to include those who would and would not be considered to require ongoing care in acute hospital. SNCT level 1 was included in the definition of eR2R in two ways, including (eR2Rab) and excluding (eR2Ra) level 1b, to determine if this affected performance.

The primary analysis of eR2R was for patients who had been in hospital for more than 24 hours at midnight. Discharge over the course of the subsequent 24 hours was evaluated. Secondary analyses were undertaken for the set of patients in a bed at 08:00 and at 16:00 to define any change in eR2R performance in these different cross-sections of the in-patient population. Three calendar years were analysed separately to assess the effects of the COVID-19 pandemic.

## Statistics

Initially, daily numbers of patients with R2R quantified both as absolute numbers and a proportion of the total number of beds were plotted for national centres and used to calculate between-centre and within-centre variation. These data are analysed as beds occupied at the specified time of day, where the bed inherits the demographics, comorbidities and other qualities of the occupying patient. This represents the in-patient population in cross-section.

For the local analysis of eR2R: the term patient-day was used to refer to a bed with the qualities of the occupying patient at the time of the analysis. The in-patient population is described as means of patient-days thereby representing a cross-section of the group. The performance of eR2R as a predictor of remaining in hospital (or absence of eR2R as a predictor of discharge) was reported as a true positive rate (TPR) and true negative rate (TNR), positive predictive value (PPV), negative predictive value (NPV) and Youden's J statistic (TPR+TNR-1), where positive is remains in hospital and negative is discharge from hospital within 24 hours.

Normally distributed variables are reported as arithmetic means±SD, with medians and ranges used otherwise. Between-centre variation was assessed by analysis of variance. This included a model accounting for day of the week as a fixed effect and the centre as a random effect. All analyses were performed using IBM SPSS V.22 (IBM Corp), with $p<0.05$ deemed to be indicative of statistical significance throughout.

### Patient and public involvement

The research question and topic were agreed following patient/public discussion groups about NHSE discharge policies. Patients/public reviewed the data fields included in the study, with the PIONEER Data Trust Committee providing support for the project (a group of patient/public members who review studies using health data[9]). A patient/public group has reviewed the results and has written a lay summary for study dissemination to patient groups.

## RESULTS

### R2R reporting in England, November 20–February 21

Across 10 164 available centre-days, accounting for 6 602 706 patient-days, the number of patients reported without R2R as a proportion of in-patients varied significantly between centres ($p<0.0001$). Individual centre means ranged from 6.7%±2.5% to 59.9%±13.8% (figure 1A). There was also marked within-centre variation (figure 1A), with coefficients of variation (CV) ranging from 8.2% up to 59.3%. Of patients not meeting R2R criteria, the proportion discharged over the following 24 hours, varied significantly between centres ($p<0.0001$). Individual centre means ranged from 14.0%±7.4% to 85.8%±25.2% (figure 1B). There was also marked within centre variation, with CV ranging from 6.4% up to 83.2%. These data are shown as median and IQR in online supplemental figure S1A,S1B). The proportion of patients without R2R and the proportion of that group discharged within 24 hours were only weakly correlated ($R^2=0.12$; Online supplemental figure S2).

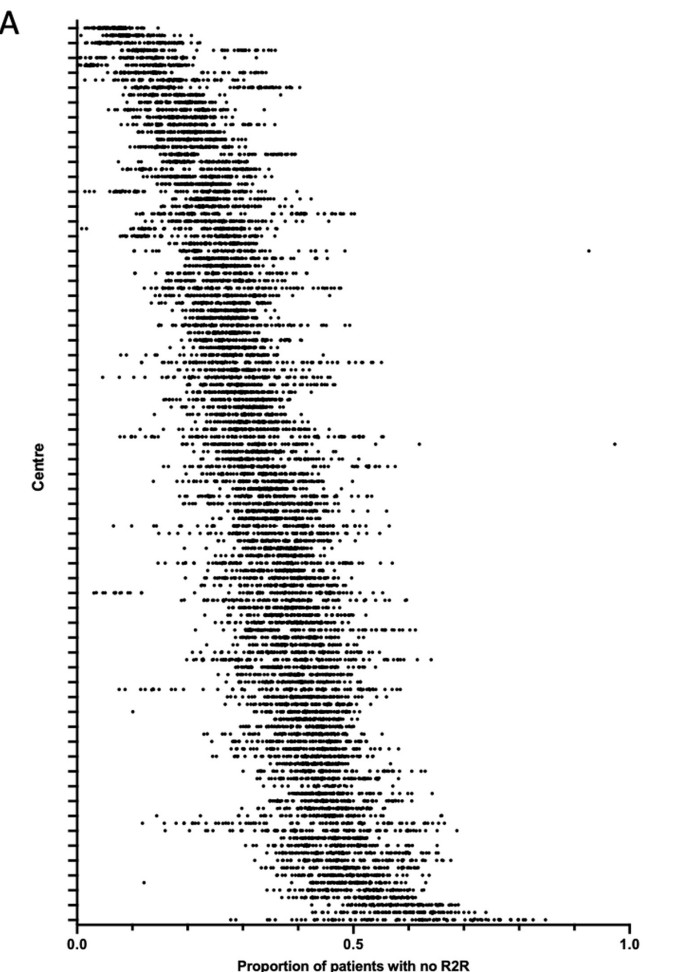

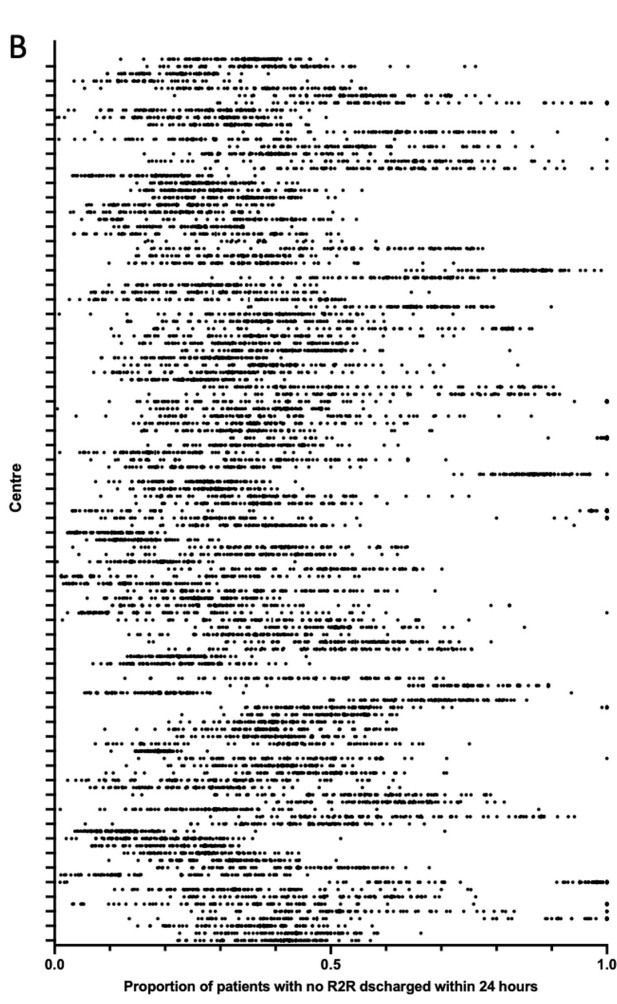

**Figure 1** National reporting of R2R criteria. The proportion of patients with no R2R (A) and of that group the proportion of patients discharged within 24 hours (B) reported to Strategic Data Collection Service (SDCS) NHS Digital, UK from 29 November 2021 to 20 February 2022 across 121 centres. Each dot represents result for a single centre-day. We have ordered centres in both A and B according to the median value of proportion of patients with R2R (see online supplemental figure S3 for median and IQRs). R2R, reason to reside.

**Table 3** Demographics of patients meeting and not meeting R2R criteria on presentation to QEHB in the censor period

|  | All QEHB patient days | Meeting eR2Rab | Not meeting eR2Rab |
|---|---|---|---|
| N | 1 039 592 | 919 751 (88.5) | 119 841 (11.5) |
| Age in years*: median (IQR) | 68 (53–80) | 69 (54–81) | 63 (48–76) |
| Sex* (n, %) |  |  |  |
| Female | 488 120 (47.0) | 434 418 (47.2) | 53 702 (44.8) |
| Male | 546 061 (52.5) | 484 816 (52.7) | 61 245 (51.1) |
| Not recorded | 5411 (0.5) | 517 (0.1) | 4894 (4.1) |
| Self-reported ethnicity* (n, %) |  |  |  |
| White | 784 528 (75.5) | 698 573 (76.0) | 85 955 (71.7) |
| Mixed/multiple | 12 983 (1.2) | 11 023 (1.2) | 1960 (1.6) |
| South Asian/Asian British | 114 049 (11.0) | 98 903 (10.8) | 15 146 (12.6) |
| Black/African/Caribbean/black British | 51 122 (4.9) | 43 991 (4.8) | 7131 (6.0) |
| Other ethnic group | 19 475 (1.9) | 16 623 (1.8) | 2852 (2.4) |
| Not known | 57 435 (5.5) | 50 638 (5.5) | 6797 (5.7) |
| Co-morbidity count* (n, %) |  |  |  |
| None | 196 121 (18.9) | 164 704 (17.9) | 31 417 (26.2) |
| 1–2 | 474 922 (45.7) | 423 200 (46.0) | 51 722 (43.2) |
| 3 or more | 368 549 (35.5) | 331 847 (36.1) | 36 702 (30.6) |
| Morbidities (n, %) |  |  |  |
| Hypertension* | 492 160 (47.3) | 439 930 (47.8) | 52 230 (43.6) |
| Cerebrovascular disease* | 159 316 (15.3) | 147 676 (16.1) | 11 640 (9.7) |
| Atrial fibrillation* | 224 501 (21.6) | 204 458 (22.2) | 20 043 (16.7) |
| Ischaemic heart disease, angina, myocardial infarct* | 198 480 (19.1) | 173 708 (18.9) | 24 772 (20.7) |
| Diabetes (types 1 and 2)* | 271 505 (26.1) | 242 328 (26.3) | 29 177 (24.3) |
| Asthma* | 103 679 (10.0) | 91 136 (9.9) | 12 543 (10.5) |
| COPD* | 112 731 (10.8) | 103 882 (11.3) | 8849 (7.4) |
| Interstitial lung disease* | 2533 (0.2) | 2380 (0.3) | 153 (0.1) |
| Chronic kidney disease* | 198 052 (19.1) | 178 284 (19.4) | 19 768 (16.5) |
| Any active malignancy* | 215 959 (20.8) | 194 419 (21.1) | 21 540 (18.0) |
| Dementia (all types)* | 65 272 (6.3) | 61 324 (6.7) | 3948 (3.3) |
| English Indices of deprivation |  |  |  |
| 1 | 430 114 (41.4) | 382 132 (41.5) | 47 982 (40.0) |
| 2 | 222 478 (21.4) | 197 999 (21.5) | 24 479 (20.4) |
| 3 | 178 565 (17.2) | 158 047 (17.2) | 20 518 (17.1) |
| 4 | 107 747 (10.4) | 96 115 (10.5) | 11 632 (9.7) |
| 5 | 75 854 (7.3) | 67 296 (7.3) | 8558 (7.1) |
| Not recorded | 24 834 (2.4) | 18 162 (2.0) | 6672 (5.6) |
| Care escalation to ITU (n, %) | 101 017 (9.7) | 93 080 (10.1) | 7937 (6.6) |

Data are number (percentage) of patients in a bed at 00:00. Ethnicity was self-reported. Medical conditions were physician confirmed and checked against admission and linked primary care notes. English indices of deprivation were calculated using postcode.
*Significant difference between meeting and not meeting eR2Rab (p<0.05 in univariate analysis).
eR2R, electronic R2R; R2R, reason to reside.

## Performance of eR2R at QEHB

Standardised definitions corresponding to the elements of R2R (table 2) were used to analyse data from QEHB, on 1 214 480 in-patient days, between 1 January 2019 and 31 December 2021. The demographic and clinical details of that population are summarised in table 3, which also shows that those meeting the definition of eR2Rab were older and more likely to have one or more comorbidities than those who did not. Variation in the daily number of patients with or without an eR2R is shown in online supplemental figure S3.

## Criteria contributing to eR2R

Given the potential for the COVID-19 pandemic to affect R2R, calendar years were analysed separately. The number of patients meeting any given eR2R criterion are shown in table 4A. The progressive contribution of different elements of the definition of eR2R assessed daily in a modified

**Table 4** (A) The number (percentage) of patient-days on which each eR2R data definition was met. (B) A phased analysis undertaken for each day and presented as a modified Consort diagram

| (A) The number (percentage) of patient-days on which each eR2R data definition was met | | | |
|---|---|---|---|
| Year | 2019 | 2020 | 2021 |
| Criterion | n (%) | n (%) | n (%) |
| ICU | 22 899 (6.1) | 20 326 (6.7) | 21 305 |
| TAP surgery 72 hours | 3783 (1.0) | 3010 (1.0) | 3974 |
| Lower limb surgery 48 hours | 285 (0.1) | 252 (0.1) | 221 (0.1) |
| Invasive surgery 24 hours | 1861 (0.5) | 1613 (0.5) | 1988 (0.6) |
| NEWS2>3 24 hours | 93 501 (24.8) | 85 123 (27.9) | 97 722 (27.3) |
| O2 treatment 24 hours | 77 949 (20.7) | 69 355 (22.7) | 77 202 (21.6) |
| Insulin infusion 24 hours | 10 951 (2.9) | 10 860 (3.6) | 12 496 (3.5) |
| Intravenous fluids 24 hours | 79 802 (21.2) | 71 376 (23.4) | 80 246 (22.4) |
| Intravenous medication administered in last 24 hours>=three times a day | 95 034 (25.2) | 81 174 (26.6) | 91 573 (25.6) |
| Intravenous medication prescribed in last 24 hours>=three times a day | 21 543 (5.7) | 17 866 (5.9) | 19 249 (5.4) |
| SNCT dependency 1a, 2, 3 | 99 139 (26.3) | 72 226 (23.7) | 88 832 (54.8) |
| COMA score <=12 in last 24 hours | 6594 (1.8) | 6448 (2.1) | 6664 (1.9) |
| End of Life care definition met in last 24 hours | 5359 (1.4) | 4747 (1.6) | 5075 (1.4) |
| SNCT dependency 1b | 172 659 (45.8) | 160 380 (52.5) | 179 527 (50.2) |
| Total number of patient days | 376 684 | 305 254 | 357 654 |
| (B) A phased analysis undertaken for each day and presented as a modified Consort diagram | | | |
| Year | 2019 | 2020 | 2021 |
| Criterion | Mean % (SD) | Mean % (SD) | Mean % (SD) |
| ICU | 6.1% (0.44) | 7.1% (3.10) | 6.0% (2.16) |
| TAP surgery 72 hours | 0.7% (0.35) | 0.7% (0.37) | 0.8% (0.45) |
| Lower limb surgery 48 hours | 0.1% (0.07) | 0.1% (0.11) | 0.1% (0.08) |
| Invasive surgery 24 hours | 0.2% (0.15) | 0.2% (0.18) | 0.2% (0.15) |
| NEWS2>3 24 hours | 24.2% (2.28) | 27.5% (3.82) | 26.6% (3.64) |
| O2 treatment 24 hours | 4.0% (0.61) | 3.9% (0.72) | 3.6% (0.68) |
| Insulin infusion 24 hours | 0.5% (0.24) | 0.6% (0.28) | 0.5% (0.23) |
| Intravenous fluids 24 hours | 8.8% (1.09) | 9.5% (1.37) | 9.6% (1.24) |
| Intravenous medication admin 24 hours>=three times a day | 7.7% (1.05) | 7.4% (1.29) | 7.5% (1.17) |
| Intravenous medication prescribed 24 hours | 0.7% (0.28) | 0.6% (0.29) | 0.6% (0.27) |
| SNCT dependency 1a, 2, 3 | 8.8% (1.42) | 6.7% (1.21) | 7.8% (1.12) |
| COMA score<=12 in the last 24 hours | 0.0% (0.05) | 0.0% (0.08) | 0.0% (0.06) |
| End of life 24 hours | 0.5% (0.24) | 0.4% (0.27) | 0.4% (0.19) |
| SNCT dependency 1b | 24.5% (1.88) | 25.5% (3.53) | 25.3% (2.59) |
| No eR2Rab total | 13.3% (1.50) | 9.8% (2.29) | 10.9% (1.87) |
| No eR2Ra total | 37.8% (2.38) | 35.3% (5.08) | 36.2% (3.60) |

The number (percentage) of patient days on which each eR2R definition was met. The population was in-patients at 24.00 with length of stay ≥24 hours.
The progressive contribution of each element to the definition of eR2R was calculated as proportion of the whole population. These were aggregated by calendar year. The order of the phased analysis was determined by the researchers to be that which was most informative, and which placed objective definitions earlier. SNCT dependency is a global nursing assessment and therefore was placed last.
eR2R, electronic R2R; ICU, intensive care unit; R2R, reason to reside; SNCT, Safer Nursing Care Tool.

Consort table are summarised in table 4B. The proportion of patients not meeting eR2R criteria exhibited relatively little day-to-day variation in 2019 (eR2Rab, CV=11.2%; eR2Ra, CV=6.3%), although somewhat higher in the context of case mix variation consequent on peaks of patients admitted with COVID-19 in 2020 (eR2Rab, CV=23.3%; eR2Ra, CV=14.4%) and 2021 (eR2Rab, CV=17.1%; eR2Ra, CV=9.9%). The criteria contributing most to eR2R status included acuity level (NEWS2>3), SNCT level nursing requirement, being on intensive care and requiring intravenous medications or fluids.

### Informedness of eR2R for discharge in the next 24 hours
For the outcome discharge (remain −)/no discharge (remain +) within 24 hours, across the three different years, the eR2Ra TPR lay between 0.63 and 0.65, TNR between 0.46 and 0.47, the PPV was 0.91 and NPV between 0.12 and 0.15; the eR2Rab TPR lay between 0.88 and 0.91, TNR between 0.18 and 0.24,

**Table 5** Contingency tables showing the number of patients meeting criteria for (A) eR2Ra and (B) eR2ab

| (A) | | | | | (B) | | | | |
|---|---|---|---|---|---|---|---|---|---|
| | | **Remain** | | | | | **Remain** | | |
| | **2019** | **Yes (+)** | **No (−)** | **Total** | | **2019** | **Yes (+)** | **No (−)** | **Total** |
| eR2Ra | Yes (+) | 213 382 | 20 845 | 234 227 | eR2Rab | Yes (+) | 297 172 | 29 372 | 326 544 |
| | No (−) | 124 874 | 17 583 | 142 457 | | No (−) | 41 084 | 9056 | 50 140 |
| | Total | 338 256 | 38 428 | 376 684 | | Total | 338 256 | 38 428 | 376 684 |
| | | **Remain** | | | | | **Remain** | | |
| | **2020** | **Yes (+)** | **No (−)** | **Total** | | **2020** | **Yes (+)** | **No (−)** | **Total** |
| eR2Ra | Yes (+) | 177 065 | 18 292 | 195 357 | eR2Rab | Yes (+) | 246 461 | 28 026 | 274 487 |
| | No (−) | 93 947 | 15 950 | 109 897 | | No (−) | 24 551 | 6216 | 30 767 |
| | Total | 271 012 | 34 242 | 305 254 | | Total | 271 012 | 34 242 | 305 254 |
| | | **Remain** | | | | | **Remain** | | |
| | **2021** | **Yes (+)** | **No (−)** | **Total** | | **2021** | **Yes (+)** | **No (−)** | **Total** |
| eR2Ra | Yes (+) | 208 068 | 20 084 | 228 152 | **eR2Rab** | Yes (+) | 288 384 | 30 336 | 318 720 |
| | No (−) | 112 007 | 17 495 | 129 502 | | No (−) | 31 691 | 7243 | 38 934 |
| | Total | 320 075 | 37 579 | 357 654 | | Total | 320 075 | 37 579 | 357 654 |

The tables show numbers of patients meeting R2R criteria and the corresponding number of patients who remain in hospital over the next 24 hours or do not (were discharged), for the in-patient population at 00:00. For eR2Ra, the TPR varied between 0.62–0.65 and TNR 0.46–0.51, across three different years and three different time points. For eR2Rab, the TPR varied between 0.87–0.91 and TNR 0.18–0.25, across three different years and three different time points. Online supplemental table S3 shows the same data for the in-patient population at 16:00. See online supplemental table S3 for all sensitivity and specificity analysis.
ER2R, electronic R2R; R2R, reason to reside.

the PPV between 0.90 and 0.91 and NPV between 0.18 and 0.20 (table 5). The J statistic for both definitions lay between 0.09 and 0.12. In secondary analyses based on the in-patient population at 08.00 and at 16.00 the J-statistic ranged between 0.10–0.14 and 0.10–0.15, respectively (online supplemental table S3A,S3B).

### In-patients not meeting eR2R

The demographic and clinical details of patient who did not meet the eR2Rab definition, stratified by discharge in the subsequent 24 hours, are shown in online supplemental table S5. For patient-days on which discharge occurred within 24 hours, there was significantly higher representation of those with no documented comorbidities 29.2% vs 24.0% (p<0.0001). In those that remained in hospital, 61.2% met eR2R criteria on subsequent days (76% within the next 24 hours). Of all those that remained, 21.9% acquired a NEWS2>3, 32.8% received iv fluids or drugs >3 times/day and 1.9% were admitted to ICU.

### DISCUSSION

Assessment of an individual patient's R2R has been promoted as a tool to improve the identification of those who could be discharged from acute hospitals in England. The proportion of in-patients with R2R and their rate of discharge has then been used to evaluate the operational efficiency of acute hospitals and their adjacent health and social care system.[1 10] This paper presents findings

to suggest that as currently constituted, R2R is of limited value for these purposes.

The high levels of variation in R2R-related metrics, within and between centres in England, has been attributed to variation in case mix and operational efficiency.[11] However, such extremes of variation are not observed in other metrics that use established data standards. Furthermore, the proportion of patients not meeting R2R criteria correlates poorly with their rate of discharge over the subsequent 24 hours, whereas one might anticipate that such closely related measures of operational efficiency would reflect one another. These findings are most obviously accounted for by the fact that R2R does not constitute a semantic data model. It is therefore susceptible to differing interpretation by individuals and centres. This applies to all the concepts described by R2R, but most obviously those that are necessarily subjective, such as 'acute functional impairment in excess of home/community care provision' and 'diminished level of consciousness where recovery is realistic'.[12 13]

We therefore developed machine readable data definitions corresponding to each concept, allowing consistent analysis of R2R at scale, using data derived from the EHR in our centre. The SNCT is a global nursing assessment of acuity and dependency that was developed to guide workforce deployment. It is regularly recorded within the EHR at our centre. Because level 1b describes a group of patients who are highly dependent on nursing care for daily activities, this was mapped onto the R2R concept

'acute functional impairment in excess of home/community care provision'. However, since the definition of level 1b could include a group of patients suitable for discharge to a less acute setting, two definitions or eR2R were tested, with and without SNCT 1b. Our analysis is therefore likely to represent two extremes of inclusion of patients with acute functional impairment.

Within centre variation in eR2R was low, consistent with it minimising individual interpretation of each data element. eR2R was a poor predictor of discharge within 24 hours.[14] Youden's Index was consistently <0.15 across three calendar years, three different times of day and two eR2R definitions. For a dichotomous test such as eR2R, a Youden's Index >0.50 is generally considered the empirical benchmark for a test to support clinical decision making.[15] eR2R is therefore unsuited to the provision of clinical decision support tool for discharge. It does not define a subpopulation on which to assess discharge performance.[16] The limitations of R2R are not entirely surprising, given the need to interpret concepts that are not semantically defined. Although addressed by eR2R, it nevertheless remains a simple series of binary responses to questions that have not been validated for the purpose of discharge prediction. For example, NEWS2 was validated as an acuity score to quantify physiological instability on initial presentation to hospital.[17] It was not developed and has not been validated, as a triage tool to assess fitness to leave hospital, at any threshold.

Importantly, more than half of those who remain in hospital without eR2R, subsequently acquired eR2R. This group of patients were older and had multiple long-term health conditions, suggesting that there were clinical grounds for that decision, although undefined. This subpopulation requires further study.

There are limitations to our analysis. The eR2R was assessed in only one centre, although one that serves a diverse, multiethnic, urban population, in which more than 1.2 million patient days were assessed. Patients admitted for <24 hours at the time of analysis were excluded to allow clinical decisions to be made and executed. The first day postadmission is a highly dynamic situation, with frequent clinical review, a setting in which this embodiment of clinical decision support is arguably less relevant. Another more intrinsic problem is that there is no gold standard by which to define all patients suitable for discharge so that actual discharge was used as a fair test when evaluating the performance of eR2R.[18] This assumes that patients actually discharged are part of a continuous population of all those who could be discharged. It is also the case that each R2R element could be defined or operationalised in slightly different ways by healthcare professionals when being applied in clinical settings. Our data analysis, with clear definitions for each parameter within the eR2R does not include the 'art' of clinical medicine but does enable consistent comparisons to be made across time and localities.

It is important to validate and evaluate tests within their intended setting. The effects of embedding new care pathways or tools within clinical service delivery, without appropriate evaluation, are increasingly described. There is significant opportunity for unintended consequences to arise from the implementation of poorly considered clinical decision support,[19] particularly when there is competition for clinical resource. This has been recently discussed for NEWS2,[20] sepsis alerting and COVID-19 virtual wards.[21] R2R has been endorsed and adopted but without validation or consideration of the unintended consequences of its application. This is not to contend that a significant number of in-patients could not be discharged earlier, simply that there is no evidence that R2R can support clinical decision making. The collective limitations of R2R identified are likely to account for variation in nationally reported metrics that are difficult to explain.

Our study highlights the need for reproducible standardised data definitions to support both implementation and validation of any tool that purports to support clinical decision making. Further research should focus on building, validating and refining tools to inform clinical decisions.

**Author affiliations**
[1]PIONEER Data Hub, University of Birmingham, Birmingham, UK
[2]Department of Acute Medicine, University Hospitals Birmingham NHS Foundation Trust, Birmingham, UK
[3]Department of Research Informatics, University Hospitals Birmingham NHS Foundation Trust, Birmingham, UK
[4]Renal Medicine, University Hospitals Birmingham NHS Foundation Trust, Birmingham, West Midlands, UK
[5]Better Care Programme and Midlands Site, HDR UK, Birmingham, West Midlands, UK

**Acknowledgements** This work was supported by the PIONEER Data Hub (see www.pioneerdatahub.co.uk) and by the PIONEER patient and public advisory group and Data Trust Committee.

**Contributors** SB and ES conceived the study; SG, FE, DM and KR conducted data analysis. ES and SG wrote the first draft of the study. All authors contributed to the study manuscript. SB is senior author and manuscript guarantor.

**Funding** This work was funded by Health Data Research - UK (PIONEER2019).

**Competing interests** FE, DM and KR have no relevant conflicts of interest. SG reports grant funding from HDR-UK. ES reports grant funding from HDR UK, Innovate UK, MRC, NIHR, British Lung Foundation and Alpha 1 Foundation. SB reports funding from HDR-UK.

**Patient and public involvement** Patients and/or the public were involved in the design, or conduct, or reporting, or dissemination plans of this research. Refer to the Methods section for further details.

**Patient consent for publication** Not applicable.

**Ethics approval** This study used unconsented, anonymous health data, and all study activity was approved by the East Midlands–Derby REC (reference: 20/EM/0158). Specific approvals were provided by East Midlands–Derby REC (reference: 20/EM/0158) to use unconsented, anonymised health data.

**Provenance and peer review** Not commissioned; externally peer reviewed.

**Data availability statement** Data are available on reasonable request. The anonymised dataset used for analysis is available on reasonable request from the PIONEER Data Hub on submission of a data request form, see www.pioneerdatahub.co.uk for a copy of the form and processes for data access.

responsibility arising from any reliance placed on the content. Where the content includes any translated material, BMJ does not warrant the accuracy and reliability of the translations (including but not limited to local regulations, clinical guidelines, terminology, drug names and drug dosages), and is not responsible for any error and/or omissions arising from translation and adaptation or otherwise.

**ORCID iDs**
Elizabeth Sapey http://orcid.org/0000-0003-3454-5482
Felicity Evison http://orcid.org/0000-0002-9378-7548
Simon Ball http://orcid.org/0000-0001-7410-5268

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
