## [Reviewer comments · BMJ Open]

ARTICLE DETAILS

TITLE (PROVISIONAL)	Variability and performance of NHS England's 'Reason to Reside' criteria in predicting hospital discharge in acute hospitals in England: a retrospective, observational cohort study
AUTHORS	Sapey, Elizabeth; Gallier, Suzy; Evison, Felicity; McNulty, David; Reeves, Katherine; Ball, Simon

VERSION 1 – REVIEW

REVIEWER	Tim Cooksley Manchester University Foundation Trust
REVIEW RETURNED	03-Sep-2022

GENERAL COMMENTS	The authors conduct an eloquent analysis of the recently published reason to reside guidance in response to the COVID19 pandemic. There was poor predictive value of the criteria to predict discharge within 24 hours and high levels of inter-centre variability. The intra-centre variability reflects recent SAMBA data of acute medical admissions which also showed significant variation between Trusts in the proportion of acute medical attendances admitted overnight. (Atkin C et al. Length of stay in Acute Medical Admissions: Analysis from the Society for Acute Medicine Benchmarking Audit. Acute Med. 2022;21(1):27-33. The need for improved, reproducible and standardised definitions for acute care metrics are an essential area for future research as the authors outline. This is an important paper demonstrate both the need and challenges of doing this. This is an interesting and important addition to the evidence in this field.
--

REVIEWER	Hiroyuki Ohbe The University of Tokyo
REVIEW RETURNED	07-Nov-2022

GENERAL COMMENTS	The manuscript "The variability and performance of NHS England's "Reason to Reside" criteria in predicting hospital discharge in acute hospitals in England. An observational study." is a retrospective observational cohort study using routinely collected health data in acute hospitals in England. There is currently no gold standard for discharge criteria, and the R2R in the UK is very interesting. The authors' evaluation of R2R is also very important. The result that R2R performs poorly and is not expected to contribute to discharge planning provides new insights to the medical community. I have several minor comments to improve this paper. Methods. I may have missed it, but please provide a detailed definition of "patients reported without R2R". Please specify that patients reported without R2R are patients who do not fit all of the
--

	metrics. Please also indicate who measures the R2R, how often the R2R is measured, and how often it is missing. Figures 1A and 1B are difficult to understand. I recommend that the authors add means and confidence intervals and rearrange the figures in order of mean from the top to the bottom to show the variation among hospitals. If I am not mistaken, eR2R was only measured at Queen Elizabeth Hospital Birmingham hospital. So the inter-centre variability mentioned in Page 3 Lines 24-26 is not assessable. The J statistic is unfamiliar. Why did the authors not use sensitivity and specificity or AUROC?
--	---

VERSION 1 – AUTHOR RESPONSE

Reviewer: 1

Dr. Tim Cooksley, Manchester University Foundation Trust:

The authors conduct an eloquent analysis of the recently published reason to reside guidance in response to the COVID19 pandemic.

There was poor predictive value of the criteria to predict discharge within 24 hours and high levels of inter-centre variability. The intra-centre variability reflects recent SAMBA data of acute medical admissions which also showed significant variation between Trusts in the proportion of acute medical attendances admitted overnight. (Atkin C et al. Length of stay in Acute Medical Admissions: Analysis from the Society for Acute Medicine Benchmarking Audit. *Acute Med.* 2022;21(1):27-33.

The need for improved, reproducible and standardised definitions for acute care metrics are an essential area for future research as the author’s outline. This is an important paper demonstrate both the need and challenges of doing this. This is an interesting and important addition to the evidence in this field.

Response: Thank you for your kind comments.

Reviewer: 2

Dr. Hiroyuki Ohbe, The University of Tokyo:

The manuscript “The variability and performance of NHS England’s “Reason to Reside” criteria in predicting hospital discharge in acute hospitals in England. An observational study.” is a retrospective observational cohort study using routinely collected health data in acute hospitals in England. There is currently no gold standard for discharge criteria, and the R2R in the UK is very interesting. The authors’ evaluation of R2R is also very important. The result that R2R performs poorly and is not expected to contribute to discharge planning provides new insights to the medical community. I have several minor comments to improve this paper.

Response: Thank you for these kind comments.

Comment. Methods. I may have missed it, but please provide a detailed definition of “patients reported without R2R”. Please specify that patients reported without R2R are patients who do not fit all of the metrics. Please also indicate who measures the R2R, how often the R2R is measured, and how often it is missing.

Response:

Thank you for this question. We have reworded the methodology to clarity in our description. To answer directly, the NHS definition is that in reference 1 and referred to in Table 1 legend. As the reviewer will appreciate, our paper is in direct response to our concerns regarding the lack of a precise, interoperable definition of R2R.

We have added the following sentences to the methods section:

“The R2R criteria are as described(1), and are also provided in the legend of Table 1“

“The number of patients with no right to reside were submitted once a day by each NHS trust, based upon local interpretation of the definition in reference 1. This required none of the criteria to be met at the time of local data collection.”

We have also added an additional sentence to the legend of Table 1

“Table 1

Legend. The policy and operating model for hospital discharge and community support within the National Health Service in England states that every person on every general ward should be reviewed on a twice daily ward round to determine whether they meet R2R. If the answer to each question is ‘no’, the policy states that active consideration for discharge to a less acute setting must be made (1). In daily data returns, the number of patients to whom this applied were counted at a single, locally defined, time point.”

We cannot comment on the influence or handling of missing data at an individual patient-day level across England. NHSE required that all patients were classified as having a reason to reside, otherwise they were considered as not having a reason to reside (there was no unknown option in returns to NHSE).

We believe the following sentence in the methods section deals with missing national data returns:

'This comprised instances where R2R data were not recorded (N=184 data points); where the total numbers of beds were either zero, missing, or clearly spurious (N=37 data points); or where there were more patients with no R2R than the total number of beds (N=3 data points). The national data are shown for the other N=121 centres, excluding UHB.'

Comment. Figures 1A and 1B are difficult to understand. I recommend that the authors add means and confidence intervals and rearrange the figures in order of mean from the top to the bottom to show the variation among hospitals.

Response: Thank you for this comment. We think that it makes our point perfectly: that it is not possible to meaningfully interpret the results of R2R. The degree of intra and inter centre variation is so high as to be operationally implausible given the relative stability of other nationally reported metrics such as bed occupancy, discharge rates etc. The lack of any standardised methodology for its collection and reporting is the likely to be the cause of this degree of variation. This relates to the point above.

We have reordered Fig 1a as requested, however point out that this creates apparent structure without an a priori hypothesis to determine that order or indeed any apparent pattern in post-hoc analysis (for example based upon geography, hospital size etc). We did not want to lose the correspondence of centres across the rows in Fig 1a and 1b so that the order of 1b is determined by 1a. (The order originally reported was simply that in which NHS statistical tables ordered on centre codes). We have also changed the legend to account for this, saying

'Legend ... We have ordered centres in both Figure 1 a and 1 b according to the median value of proportion of patients with R2R. (See Fig S3 for medians and interquartile ranges)'

We discuss the range of means and standard deviations in our results in the context of the very high levels of inter- and intra- centre variability:

'The number of patients reported without R2R as a proportion of in-patients, varied significantly between centres ($p < 0.0001$). Individual centre means ranged from $6.7\% \pm 2.5\%$ to $59.9\% \pm 13.8\%$ (Figure 1a). There was also marked within centre variation (Figure 1a), with coefficients of variation (CV) ranging from 8.2% up to 59.3%.

Of patients not meeting R2R criteria, the proportion discharged over the following 24 hours, varied significantly between centres ($p < 0.0001$). Individual centre means ranged from $14.0\% \pm 7.4\%$ to $85.8\% \pm 25.2\%$ (Figure 1b). There was also marked within centre variation, with coefficients of variation ranging from 6.4% up to 83.2%'

When discussed with a number of senior statisticians in the UK, their strongly held view was that presentation of the raw data was so compelling that alternative representations such as box and whiskers for each centre, detracted from Fig 1. As such we have not modified the diagram further, but we have included a second version of this figure showing the median and IQR across the centres as a supplementary figure (Fig S3). We make reference to the supplementary median and IQR plot in the legend.

Comment. If I am not mistaken, eR2R was only measured at Queen Elizabeth Hospital Birmingham hospital. So the inter-centre variability mentioned in Page 3 Lines 24-26 is not assessable.

Response: Thank you. That sentence in fact relates to R2R calculated from national data, as described in the methods section “Initially, daily numbers of patients with R2R quantified both as absolute numbers and a proportion of the total number of beds, were plotted for national centres and used to calculate between-centre and within-centre variation.” We have tried to make this clearer.

Comment. The J statistic is unfamiliar. Why did the authors not use sensitivity and specificity or AUROC?

Response. Youden's J statistic is a single statistic that captures the performance of a dichotomous diagnostic test.

Index for rating diagnostic tests, W. J. Youden, Cancer 1950 3 (1) 32-35

[https://doi.org/10.1002/1097-0142\(1950\)3:1<32::AID-CNCR2820030106>3.0.CO;2-3](https://doi.org/10.1002/1097-0142(1950)3:1<32::AID-CNCR2820030106>3.0.CO;2-3)

$J = \text{sensitivity} + \text{specificity} - 1$

For a diagnostic test that is progressive rather than dichotomous the J statistic = height of the ROC curve above the line of equivalence at any given threshold, as illustrated below.

However, as the reviewer will appreciate the R2R is a dichotomous test, so no ROC curve can be generated. In this case the J statistic is an appropriate measure of performance.

Although we could quote the sensitivity and specificity separately, when the J statistic is as low as observed, test performance is close to being random. That being the case reporting sensitivity and specificity separately ceases to add value. It is for that reason that we focused our reporting on the J statistic rather than separate representation of sensitivity and specificity.

We refer to the range of sensitivity and specificity in legend to Table 5.

In order to address the reviewers' concerns, we have added a supplementary table (Table S4):

'Sensitivity, Specificity and J statistic calculations for Table 5

Contingency tables showing the number of patients meeting criteria for eR2Ra (A) and eR2ab(B)'

	eR2Ra			eR2Rab		
Year	2019	2020	2021	2019	2020	2021
Sensitivity	0.63	0.65	0.65	0.88	0.91	0.90
Specificity	0.46	0.47	0.47	0.24	0.18	0.19
J statistic	9%	12%	12%	11%	9%	9%

We also found a typographical error in Table 5 legend which has been corrected in the revised manuscript shown in red. The second 'For eR2Ra' should read 'For eR2Rab'. The labelling of the data in the tables is correct. We have also added a sentence to the legend for Table 5 referring to Table S4

Table 5. Contingency tables showing the number of patients meeting criteria for eR2Ra (A) and eR2ab(B)

Legend. The tables show numbers of patients meeting R2R criteria and the corresponding number of patients who remain in hospital over the next 24 hours or do not (were discharged), for the in-patient population at 00:00. For eR2Ra, the TPR varied between 0.62-0.65 and TNR 0.46-0.51, across 3 different years and 3 different time points. For eR2Rab, the TPR varied between 0.87-0.91 and TNR 0.18-0.25, across 3 different years and 3 different time points. Table S3 of the online supplement shows the same data for the in-patient population at 16:00. **See Table S4 of the online supplement for all sensitivity and specificity analysis.**

VERSION 2 – REVIEW

REVIEWER	Hiroyuki Ohbe The University of Tokyo
REVIEW RETURNED	07-Dec-2022
GENERAL COMMENTS	I have no further comments.